# INFERENCE TIME CAUSAL PROBING IN LLMS

## ABSTRACT

Causal probing methods aim to test and control how internal representations influence the behavior of generative models. In causal probing, an intervention modifies hidden states so that a property takes on a different value. Most existing approaches define such interventions by training an auxiliary probe classifier, which ties the method to a specific task or model and risks misalignment with the model's predictive geometry. We propose Hidden-state Driven Margin Intervention (HDMI), a probe-free, gradient-based technique that directly steers hidden states using the model's native output. HDMI applies a margin objective that increases the probability of a target continuation while decreasing that of the source, without relying on probe classifiers. We further introduce a lookahead variant (LA-HDMI) for text editing that backpropagates through the softmax embeddings, modifying the current hidden state so that the likelihood of user-specified tokens increases in next token generations while preserving fluency. To evaluate interventions, we measure completeness (whether the targeted property changes as intended) and selectivity (whether unrelated properties are preserved), and report their harmonic mean as an overall measure of reliability. HDMI consistently achieves higher reliability than prior methods on the LGD agreement corpus and the CausalGym benchmark, across Meta-Llama-3-8B-Instruct, and Pythia-70M.

## 1 INTRODUCTION

In the study of generative models, a key goal is to understand what latent properties they encode and how these properties shape generation. By property, we mean a linguistic feature of the input that the model may represent internally and that can take on distinct values. For instance, consider the sentence: "*The cats run across the yard*". Here, the subject has the property of plural number. If we change to "*The cat runs across the yard*", the property would instead be singular number.

A standard diagnostic approach, **correlational probing**, trains lightweight classifiers (commonly called "probes") on hidden states to decode a property (such as sentiment, syntactic role, or topic). These probes reveal what value of property is present in hidden states, but they do not show whether/how the model actually uses this property in predicting the next words. For example, the probe might read off "plural number" but this does not establish whether plurality is what drives the model to predict "run" instead of "runs" for the next token.

**Causal probing** addresses the aforementioned limitation by *interventions* on the hidden state and tests whether such perturbations alter the next-token distribution of the model. For instance, if we modify the hidden state so that the subject property changes from singular to plural, we would like to see whether the model accordingly modifies the verb from singular to plural form. Therefore, causal probing investigates not only what information is encoded, but also how it is used in prediction (Elazar et al., 2021; Ravfogel et al., 2020; Kumar et al., 2022). [1]

In practice, many causal probing methods rely on training a probe to obtain a direction along which to perturb the hidden state. For example, PGD (Madry et al., 2018) trains a probe to classify subject number from hidden states, and then uses the probe's gradient to adjust the hidden state such that

---

[1]Following prior causal probing work (Canby et al., 2024; Ravfogel et al., 2021; Davies et al., 2023), we sometimes refer to altered property as "counterfactuals". However, in the structural causal model framework Pearl (2009), counterfactuals require re-evaluating a system under the same randomness, whereas our setting just replaces a hidden representation with a modified one. Thus, our experiments are more accurately considered as interventional queries rather than counterfactual ones.

the perturbed state is classified by the probe with the opposite label (e.g., flipping the label from singular to plural). However, this reliance on probes introduces extra property-specific supervision and training costs, since a separate probe must be trained for each property of interest. Moreover, there is a risk of misalignment in the sense that the probe imposes its own classification boundary on the hidden state which may not coincide with how the model internally encodes and uses a property for generating next tokens.

This motivates our framing of **inference-time causal probing**. By this, we mean interventions applied entirely at inference time, requiring no probes or retraining of the generative model. Our proposed method, **Hidden-state Driven Margin Intervention (HDMI)**, directly leverages the generative model's output head as a native readout. HDMI performs a lightweight gradient-based update to hidden states that shifts the model's output distribution from the source continuation (e.g., "runs" for a singular subject) to the interventional target (e.g., "run" for a plural subject). Concretely, it uses the gradient of the logit margin (defined as the difference between the target and source token logits at the next decoding step), increasing the probability of the target being selected.

The contributions of the paper are as follows:

- Unlike prior approaches (Davies et al., 2023; Ravfogel et al., 2021; Madry et al., 2018; Goodfellow et al., 2015) that rely on probes, HDMI directly leverages the generative model head to obtain intervention directions, ensuring alignment with the model's own predictive geometry. The objective of HDMI is to maximize the logit margin that increases the target continuation's logit while decreasing the source's. This formulation naturally extends to cases where a property can be expressed by multiple acceptable continuations (e.g., plural verbs "are/were" versus singular verbs "is/was"). Moreover, the gradient computation of the logit margin reduces to a closed-form matrix–vector product, making HDMI computationally lightweight and easy to deploy (Section 4).

- We propose a "lookahead" variant of HDMI for text editing, where a user provides an edited version for a given input, and the goal is to perturb hidden states to steer generation toward the edited version while preserving fluency. Lookahead HDMI (LA-HDMI) introduces a softmax embedding transition in the token generation process and modifies the hidden state of every decoding step influenced by future edit positions (Section 5).

- Using the evaluation framework of Canby et al. (2024), compared to previous work, we show that HDMI achieves strong performance across LGD agreement and CausalGym suites. We report completeness and selectivity, metrics that show that interventions alter target properties but avoid altering unrelated properties (Section 6).

## 2 RELATED WORK

Our work is related to three research directions: (i) probing methods for analyzing the linguistic information encoded in hidden states; (ii) causal interventions on internal mechanisms in language models; and (iii) controllable generation and editing methods that steer model behavior at inference time.

**Probing.** A long line of work uses lightweight classifiers—"probes"—to read out linguistic properties from intermediate representations (Belinkov, 2022; Alain & Bengio, 2016; Hewitt & Liang, 2019; Pimentel et al., 2020). Although such probes revealed that models encode rich structure, their interpretability and causal effects on the model predictions have been debated. The fact that a property can be predicted from embedding representations (or correlated) does not imply the model relies on that property. Some methods have proposed controls to check that a probe's accuracy is not due to dataset quirks or the probe memorizing labels (Hewitt & Liang, 2019; Pimentel et al., 2020). This motivated causal probing, which intervenes on representations and measures behavioral impact.

**From correlation to causation.** Recent work shifts from correlational probing to causal analysis that intervenes on internal representations (Elazar et al., 2021; Tucker et al., 2021; Ravfogel et al., 2021; 2022; Davies et al., 2023). Concept–erasure methods attempt to remove information about attributes from representations, often to debias (Elazar & Goldberg, 2018) or perform causal analysis. Iterative Nullspace Projection (INLP) iteratively removes linearly predictive subspaces for a given

attribute from representations and projects representations onto the nullspace of linear predictors for a target attribute (Ravfogel et al., 2020; 2022). "Amnesic probing" couples such an erasure with behavioral checks to study whether removing present information about a property changes the model predictions (Elazar et al., 2021). Counterfactual interventions instead modify the hidden state so that the property changes from its factual value to a counterfactual value. Linear counterfactuals such as AlterRep push representations across rowspace hyperplanes to the counterfactual side (Ravfogel et al., 2021). Nonlinear counterfactuals include gradient-based interventions (GBIs) that optimize against an attribute probe with FGSM/PGD-style updates (Goodfellow et al., 2015; Madry et al., 2018). However, concerns persist that interventions may incompletely transform the target property or inadvertently alter non-target properties (Kumar et al., 2022; Canby et al., 2024). In contrast, our approach does not learn a probe for the intervention and instead exploits a direct, model-native gradient signal from the language model head, aligning the intervention with the model's predictive geometry at the specific decoding step.

**Controllable model behavior.** Plug-and-Play Language Models (PPLM) perform gradient-based updates to hidden states to guide the token generations (e.g., change the text topic) using external attribute classifiers, alongside KL regularization to remain on-manifold (Dathathri et al., 2019). GeDi (Krause et al., 2020) and DExperts (Liu et al., 2021) provide alternative guidance via discriminators or expert/anti-expert mixtures to make them more controllable. More recently, representation/activation engineering has explored linear directions to change the behaviors of models (Turner et al., 2023).

Our proposed approach is a form of causal probing that differs from the aforementioned work in two ways: (i) we do not require training an external attribute model or probe. HDMI uses a targeted, per-instance, per-step gradient to change a specified continuation. (ii) our margin intervention objective is to change the target property completely and not to affect unrelated properties. Our evaluation approach is akin to Canby et al. (2024), which introduced metrics for completeness (did the intervention achieve changing the value of the targeted property?) and selectivity (did it avoid altering unrelated properties?).

# 3 PROBLEM SETTING AND NOTATIONS

Let $M$ be a pretrained language model (LM) with $L$ transformer layers and vocabulary $\mathcal{V}$. For an input sequence $x_{1:T} = (x_1, \ldots, x_T)$ and a decoding step $T + 1$, we denote the layer–$\ell$ hidden representation by $h_\ell(x_{1:T}) \in \mathbb{R}^D$, where $\ell \in \{1, \ldots, L\}$, and $D$ is the model's hidden size. Unless stated otherwise, we drop the position subscript and write $h_\ell(x)$ for $h_\ell(x_{1:T})$. The model's next-token logit vector is $\phi(x_{1:T}) = W_U\, h_L(x_{1:T}) + b \in \mathbb{R}^{|\mathcal{V}|}$, where $W_U \in \mathbb{R}^{|\mathcal{V}| \times D}$ is the unembedding and $b \in \mathbb{R}^{|V|}$ is the bias. Unless stated otherwise, we drop the position subscript and write $\phi(x)$ for $\phi(x_{1:T})$ by default. The probability distribution for generating the next token is obtained via softmax,

$$P_M(\,\cdot\,|\,x_{1:T}) = \mathrm{softmax}\big(\phi(x_{1:T})\big) \in \Delta^{|\mathcal{V}|-1},$$

where $\Delta^{|\mathcal{V}|-1}$ denotes the probability simplex over $\mathcal{V}$. We assume a family of discrete latent linguistic properties $\mathcal{Z} = \{Z_1, Z_2, \ldots\}$ defined on input texts. The classical "probing" paradigm evaluates whether a hidden vector $h_\ell(x)$ correlates with a linguistic property $Z \in \mathcal{Z}$ by training a supervised classifier on hidden representations $h_\ell(x)$. Causal probing instead asks how or whether manipulating the hidden representation at layer $\ell$ changes a linguistic property $Z \in \mathcal{Z}$ and the model's next-token prediction denoted as $Y$.

Throughout the paper, we consider two types of properties. First, a *causal* property $Z_c$ that directly governs the model's next-token choice $Y$ at position $T + 1$. For instance, in English subject–verb agreement, $Z_c \in \{\text{SG}, \text{PL}\}$ encodes the subject number (singular or plural) and should impose the verb agreement at the next token prediction (e.g., $Y \in \{is, are\}$). Second, an *environment* or *nuisance* property $Z_e$ (e.g., the number of non-subject nouns in a

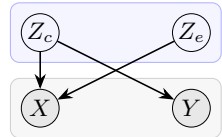

Figure 1: $X$ is the sequence $x_{1:T}$ and $Y$ is the token at $T + 1$. $Z_c$ and $Z_e$ are latent linguistic properties.

prepositional phrase) that varies with the context and does not affect $Y$ (e.g., non-subject noun number does not affect the verb, see Figure 1). Both $Z_c$ and $Z_e$ take values in finite sets $\mathcal{K}_c$ and $\mathcal{K}_e$, respectively.

Let $f_\theta : \mathbb{R}^D \to \mathbb{R}^D$ be an interventional operator:

$$\tilde{h}_\ell(x) \;=\; f_\theta\big(h_\ell(x),\, z \to z'\big), \quad \text{where } z' \neq z.$$

This operator for a given $x$, intervenes on the hidden state to change the value of $Z$ from $z$ to $z'$

# 4 Hidden-state Driven Margin Intervention (HDMI)

Let $\phi : \mathbb{R}^D \to \mathbb{R}^{|\mathcal{V}|}$ be a differentiable logit function (by default the LM head $\phi(x)$; see Section 3). Consider the input sequence $x_{1:T}$ whose next–token distribution we want to predict. Let $h_\ell(x)$ denote the hidden state representation at layer $\ell$. In the following examples, we shall denote the position $T + 1$ by [MASK].

In causal probing, the dataset provides two mutually exclusive next–token continuations or labels:

- the *source* token $v_a \in \mathcal{V}$, which is the token preferred by the model under its original distribution that realizes the factual value $z$ of $Z_c$, and
- the *target* token $v_b \in \mathcal{V}$, which realizes the counterfactual value $z'$ of $Z_c$ we wish the model to adopt.

For instance, consider the sequence "*The key [MASK] on the table.*" Without intervention, the base model might prefer $v_a = $ is at [MASK] because $z = $ SG. If we wish to flip the subject-number property to plural ($z' = $ PL), the *target* token at this step is $v_b = $ are.

Denote the indices of $v_a$ and $v_b$ in the logit vector $\phi$ by $\sigma = \text{ID}(v_a)$, and $\tau = \text{ID}(v_b)$.

**Single-token margin objective.** We aim to find a perturbed hidden state $\tilde{h}_\ell(x)$ that *raises* the logit (or log-probability) of the target token $\tau$ while *lowering* that of the source token $\sigma$ at the next decoding step. We define the margin objective as follows:

$$\mathcal{L}(x) \;=\; \phi(x)_\tau \;-\; \phi(x)_\sigma. \tag{1}$$

Maximizing this margin objective increases the probability of the target token while decreasing that of the source token. We therefore apply gradient ascent on the hidden state $h_\ell(x)$. Note that for the final layer $L$, and for the $\phi$ defined in Section 3, because $\mathcal{L}$ is linear in $h_L(x)$, computing the gradient is inexpensive:

$$\nabla_{h_L} \mathcal{L}(x) \;=\; W_U^\top \big(e_\tau - e_\sigma\big),$$

where $e_i$ is the $i$-th basis vector in $\mathbb{R}^{|\mathcal{V}|}$. For a general $\ell$: $\nabla_{h_\ell} \mathcal{L}(x) = \big(\nabla_{h_\ell} \phi(x)\big)^\top \big(e_\tau - e_\sigma\big)$, which requires computing a matrix–vector product, also a computationally lightweight calculation.

**Multi-token extension.** Sometimes the property can be realized by several acceptable tokens at the very next step. For example, for subject-verb agreement, we may wish to use both are and were for the counterfactual value $z' = $ PL, and is and was for $z = $ SG.

HDMI can optimize a set-based margin. Let $\mathcal{T}^+ = \{\tau_1, \dots, \tau_p\}$, $\mathcal{T}^- = \{\sigma_1, \dots, \sigma_q\}$, and define the loss as:

$$\mathcal{L}_{\text{set}}(x) = \sum_{i=1}^{p} \phi(x)_{\tau_i} - \sum_{j=1}^{q} \phi(x)_{\sigma_j}. \tag{2}$$

Let $u^+ = \sum_i e_{\tau_i}$ and $u^- = \sum_j e_{\sigma_j}$ so that $\mathcal{L}_{\text{set}}(x) = \phi(x)^\top (u^+ - u^-)$. For interventions at the final layer ($\ell = L$), $\nabla_{h_L} \mathcal{L}_{\text{set}}(x) = W_U^\top \big(u^+ - u^-\big)$. The gradient for general $\ell$ is $\nabla_{h_\ell} \mathcal{L}_{\text{set}}(x) = \big(\nabla_{h_\ell} \phi(x)\big)^\top \big(u^+ - u^-\big)$, which is the same as equation 1 when $p = q = 1$. Starting from the original representation $h^{(0)} = h_\ell(x)$, we perform $K$ steps of gradient ascent:

$$g^{(k)} \;=\; \nabla_{h^{(k)}} \mathcal{L}(x), \qquad h^{(k+1)} \;=\; h^{(k)} + \alpha\, g^{(k)}, \tag{3}$$

where $\alpha > 0$ is the step size. The final state representation $\tilde{h}_\ell(x) = h^{(K)}$ is substituted back into the forward pass at only layer $\ell$ and position $T + 1$ for the next token prediction; all other layers and decoding steps remain unchanged.

## 5 TEXT EDITING WITH LOOKAHEAD HDMI

In this section, we show a use–case of HDMI in which for a given input sequence $x_{1:T} = (x_1, \ldots, x_T)$, a *user* provides an edited sequence $\tilde{x}_{1:T} = (\tilde{x}_1, \ldots, \tilde{x}_T)$, and the goal is to steer the input sequence toward the edited version fluently. Rather than forcing the LM to reproduce $\tilde{x}_i$s, we *steer* the hidden states of the input sequence so that the generated sequence remains fluent while implementing the user's changes. For every decoding step $t + 1 \in \{1, \ldots, T\}$, we denote the last-token layer-$L$ hidden $h_L(x_{1:t})$ by $h_t \in \mathbb{R}^D$, the temperature by $\beta_g$, the next-token logit $\phi(x_{1:t})$ by $\phi_t$ and the next-token distribution by $y_t$. Then,

$$\phi_t = W_U h_t + b, \qquad y_t = \mathrm{softmax}(\phi_t / \beta_g).$$

Since we considered an autoregressive LM, a hidden state $h_t$, is not aware of future edits. To let the $h_t$ be influenced by *future* edit positions ($t' > t$), we must consider the transition from $h_t$ to $h_{t+1}$ through the expected embedding $m_t$ as follows:

$$h_t \rightarrow \phi_t = W_U h_t + b \rightarrow y_t = \mathrm{softmax}(\phi_t / \beta_g) \rightarrow m_t = E^\top y_t \rightarrow h_{t+1} = \mathcal{T}(m_t)$$

where $E \in \mathbb{R}^{|\mathcal{V}| \times d_e}$ is the token embedding matrix, $d_e$ is the embedding size, and $m_t \in \mathbb{R}^{d_e}$ is the expected value of the next-token prediction based on the distribution $y_t$. $\mathcal{T}$ denotes the decoder one-step transition that maps the expected next-token embedding to the next last-layer hidden state. Note that $\mathcal{T}$ maintains a cache of all past tokens' attention states, which we have omitted here for brevity. In this formulation, gradients of a margin objective (such as equation 1) can be backpropagated via a vector–Jacobian product (VJP) through the softmax–expected-embedding–transition chain to the hidden states of the past decoding steps ($t' < t$). Since we do a forward pass with $m_t$, no gradient flows through argmax/sampling, and therefore, the transitions remain differentiable. Note that autoregressive LLMs were never trained to consume combinations of expected embeddings; hence, feeding expected embeddings in their token generation process (forward pass) puts the model off-manifold and quickly collapses fluency–even without intervening on hidden states. To preserve fluency while still obtaining useful gradients, we decouple the forward token generation process from the gradient backpropagation path. At each decoding step, we choose a very low-temperature $\beta_f$ in softmax and feed that generated embedding into the model, to mimic exactly the standard inference. In parallel, to compute the gradient with backpropagation, starting from the current hidden state, we build a differentiable "lookahead" objective by using the expected embedding under a high-temperature value ($\beta_g$ near 1). In the following, we define the objective function for text editing.

Denote the set of positions that must change (specified by user edits) by $\mathcal{M} = \{ j \in \{1, \ldots, T\} : x_j \neq \tilde{x}_j \}$, and let $(a_j, b_j) = (\mathrm{ID}(x_j), \mathrm{ID}(\tilde{x}_j))$ be their respective indices. Before predicting $x_{t+1}$, we build an objective that considers the edits up to $S_{\max}$ steps ahead:

$$J_t(h_t) = \sum_{s=1}^{S_{\max}} \mathbf{1}_{\{t+s \in \mathcal{M}\}} \big[ \phi(x_{1:t+s-1})_{b_{t+s}} - \phi(x_{1:t+s-1})_{a_{t+s}} \big].$$

Maximizing this cumulative margin objective adjusts $h_t$ so that the probability of future target tokens increases while decreasing that of the source ones. To prevent deviating from the input sequence in the not edited parts and follow the input sequence logit, we also incorporate the source token logit at position $t + 1$ and regulate it using a coefficient $\lambda_{\mathrm{fact}}$:

$$J_t \leftarrow J_t + \lambda_{\mathrm{fact}} \, \phi(x_{1:t})_{a_{t+1}}, \qquad \lambda_{\mathrm{fact}} \in [0, 1].$$

Starting from $h_t^{(0)} = h_t$, we perform $K$ steps of gradient ascent

$$g_t^{(k)} = \nabla_{h_t^{(k)}} J_t, \quad h_t^{(k+1)} = h_t^{(k)} + \alpha \, g_t^{(k)},$$

where $\alpha$ is the step size. After $K$ inner steps we obtain $h_t' = h_t^{(K)}$, and form the following distribution using a low temperature $\beta_f$

$$y_t' = \mathrm{softmax}(W_U h_t' / \beta_f).$$

We show the next token by taking the argmax of $y_t'$, $x_{t+1}^\star = \arg\max y_t'$; however, we continue the forward pass with the *expected embedding* $m_t' = E^\top y_t'$.

# 6 EXPERIMENTS

## 6.1 CAUSAL PROBING WITH HDMI

We study pretrained decoder–only LMs with $L$ transformer layers. The complete code is provided in the supplementary material. Consider the input sequence $x_{1:T}$ whose next–token distribution we want to predict. Unless noted otherwise, we intervene at the final layer $\ell = L$. In this section, we often denote $h_\ell(x_{1:T})$ by $h_\ell$ for sake of brevity.

**Datasets.** We evaluate our approach on two complementary sources.

**LGD agreement corpus.** We follow the protocol of Canby et al. (2024): natural Wikipedia sentences from the original LGD dataset (Linzen et al., 2016) are filtered so that both singular and plural inflections of the target verb are in the dataset. Each dataset sample supplies (i) an input sequence ending right before the next-token prediction (the model predicts the verb denoted with [MASK] in the prompt), (ii) mutually exclusive labels $\langle v_{\text{sg}}, v_{\text{pl}} \rangle$, that we choose $v_a$ and $v_b$ from (e.g., *locks* and *lock* respectively), and (iii) annotations for the subject number $Z_c$, which is either singular (SG) or plural (PL) and, when present, the number of the most recent non-subject noun in a prepositional phrase ($Z_e$). We map $Z_c \in \{\text{SG}, \text{PL}\}$ to $\{0, 1\}$ and $Z_e$ to a 3–class label $\{0 = \varnothing, 1 = \text{SG}, 2 = \text{PL}\}$.

**CausalGym.** CausalGym (Arora et al., 2024) groups examples into *suites*, each targeting a single grammatical phenomenon (e.g., agreement with prepositional phrase (PP) distractors, subordination, clefting, filler–gap). Items are provided as *minimal pairs* $\langle x_{\text{src}}, x_{\text{cf}} \rangle$, and their corresponding labels $\langle y_{\text{src}}, y_{\text{cf}} \rangle$ that are identical except for the property of interest $Z_c$, ensuring that any difference in model preference can be attributed to this property. For instance, $\langle x_{\text{src}}, x_{\text{cf}} \rangle = \langle$*John walked because [MASK], Jane walked because [MASK]*$\rangle$ and $\langle y_{\text{src}}, y_{\text{cf}} \rangle = \langle$*he, she*$\rangle$. From each pair, we create two samples by swapping which label is active; hence, the source/target tokens (defined in Section 4) are well defined. For example, for the prompt *John walked because [MASK]*, the source token $v_a$ is *he* and the target one $v_b$ is *she*. We take the varied feature as $Z_c$ (in this example "gender") and similar to LGD dataset, we map $Z_c \in \{$*male, female*$\}$ to $\{0, 1\}$. For the environment property $Z_e$ we use a preposition–family heuristic over the prompt near the verb: $Z_e \in \{0, 1, 2, 3, 4\} \equiv \{\text{NONE}, \text{OF}, \text{IN}, \text{WITH/BY}, \text{OTHER}\}$, obtained by scanning the last 12 tokens for the most recent preposition.

A validation probe (Canby et al., 2024) is a lightweight classifier trained to *decode* a latent property from hidden states, without participating in the intervention itself. Given layer–$\ell$ representations $h_\ell(x)$, we train two probes on a disjoint split: (i) *probe*$_{Z_c}$, which estimates $P(Z_c \mid h_\ell)$ and is used to quantify **completeness** by checking whether an intervention shifts the encoding toward the counterfactual value; and (ii) *probe*$_{Z_e}$, which estimates $P(Z_e \mid h_\ell)$ and is used to quantify **selectivity** by checking that non-target properties remain stable. Importantly, validation probes are never trained on intervened representations and never see the test set; their role is to read out what the model encodes, not to learn to cope with interventions.

The data is partitioned into (1) an *interventional* split, (2) a *validation-probe* split, and (3) a held-out *test* split for two reasons. First, to avoid leakage: the interventional split is used to fit any probe-driven intervention (e.g., a $Z_c$ probe for gradient-based baselines) and to tune intervention hyper-parameters, while validation probes are fit *only* on the validation-probe split. Second, to obtain unbiased metrics: we subsample the validation-probe split so that $Z_c$ and $Z_e$ are approximately independent (Canby et al., 2024), ensuring *probe*$_{Z_c}$ cannot spuriously use $Z_e$ (and vice versa). The test split is used exclusively for final reporting of completeness, selectivity, and reliability after all training and tuning are performed.

We compared our method (HDMI) with several baselines: **HDMI (ours).** Gradient ascent on the next–token margin ( equation 1), i.e. increase $\phi(x)_\tau$ while decreasing $\phi(x)_\sigma$. In Appendix A.2, there is an ablation study on only increasing $\phi(x)_\tau$. **GBI.** Gradient–based counterfactual intervention via FGSM/PGD (Goodfellow et al., 2015; Madry et al., 2018) against the interventional $Z_c$ probe, targeted to predict a counterfactual value of $Z_c$ within an $\ell_\infty$ or $\ell_2$ ball (default: $\ell_\infty$). **AlterRep Ravfogel et al. (2021).** It modifies only the component of the representation $h_\ell$ that lies in the row space of a linear concept classifier (the span of its weight vectors) and leaves the part orthogonal to that space unchanged.

Table 1: CausalGym and LGD: Completeness, Selectivity, and Reliability for HDMI (ours), AlterRep, FGSM, and PGD on Meta-Llama-3-8B-Instruct and EleutherAI/pythia-70m. Higher value is better. C. , S., and R. are short for Completeness, Selectivity, and Reliability, respectively.

| Task | Method | Meta-Llama-3-8B-Instruct | | | EleutherAI/pythia-70m | | |
|---|---|---|---|---|---|---|---|
| | | C. | S. | R. | C. | S. | R. |
| agr_sv_num_obj-relc | HDMI | 1.0000 | 1.0000 | 1.0000 | 1.0000 | 1.0000 | 1.0000 |
| | AlterRep | 0.9800 | 1.0000 | 0.9899 | 0.9600 | 1.0000 | 0.9796 |
| | FGSM | 1.0000 | 1.0000 | 1.0000 | 1.0000 | 1.0000 | 1.0000 |
| | PGD | 0.6307 | 1.0000 | 0.7736 | 1.0000 | 1.0000 | 1.0000 |
| agr_sv_num_pp | HDMI | 1.0000 | 0.9362 | 0.9671 | 0.9984 | 0.9409 | 0.9688 |
| | AlterRep | 0.9500 | 0.8680 | 0.9072 | 0.5000 | 0.5108 | 0.5053 |
| | FGSM | 1.0000 | 0.8314 | 0.9080 | 0.9984 | 0.4906 | 0.6579 |
| | PGD | 0.3650 | 0.3648 | 0.3649 | 1.0000 | 0.8550 | 0.9218 |
| agr_refl_num_subj-relc | HDMI | 1.0000 | 1.0000 | 1.0000 | 1.0000 | 1.0000 | 1.0000 |
| | AlterRep | 0.9898 | 1.0000 | 0.9949 | 0.5000 | 1.0000 | 0.6667 |
| | FGSM | 1.0000 | 1.0000 | 1.0000 | 0.4963 | 1.0000 | 0.6634 |
| | PGD | 0.9636 | 1.0000 | 0.9815 | 0.4912 | 1.0000 | 0.6588 |
| agr_refl_num_pp | HDMI | 0.9902 | 0.8272 | 0.9014 | 1.0000 | 0.8983 | 0.9464 |
| | AlterRep | 0.9900 | 0.6540 | 0.7877 | 0.5449 | 0.8843 | 0.6743 |
| | FGSM | 0.5341 | 0.9881 | 0.6934 | 0.9642 | 0.9206 | 0.9419 |
| | PGD | 0.3335 | 0.5155 | 0.4050 | 0.6142 | 1.0000 | 0.7610 |
| gss_subord_subj-relc | HDMI | 1.0000 | 0.8969 | 0.9456 | 1.0000 | 1.0000 | 1.0000 |
| | AlterRep | 0.9700 | 0.8938 | 0.9303 | 0.6095 | 1.0000 | 0.7573 |
| | FGSM | 0.5500 | 1.0000 | 0.7097 | 0.1507 | 1.0000 | 0.2619 |
| | PGD | 0.6411 | 1.0000 | 0.7813 | 0.3463 | 1.0000 | 0.5145 |
| gss_subord_pp | HDMI | 1.0000 | 0.8969 | 0.9456 | 0.9984 | 0.7371 | 0.8481 |
| | AlterRep | 0.9700 | 0.8938 | 0.9303 | 0.6896 | 0.7051 | 0.6972 |
| | FGSM | 1.0000 | 0.9382 | 0.9681 | 0.1333 | 0.9799 | 0.2347 |
| | PGD | 0.9950 | 0.5298 | 0.6914 | 0.3015 | 0.6628 | 0.4144 |
| cleft | HDMI | 1.0000 | 0.9800 | 0.9899 | 1.0000 | 1.0000 | 1.0000 |
| | AlterRep | 0.8603 | 0.9800 | 0.9163 | 0.7300 | 1.0000 | 0.8440 |
| | FGSM | 1.0000 | 1.0000 | 1.0000 | 0.7900 | 1.0000 | 0.8827 |
| | PGD | 0.9946 | 1.0000 | 0.9973 | 0.9599 | 1.0000 | 0.9796 |
| filler_gap_hierarchy | HDMI | 1.0000 | 0.8414 | 0.9138 | 1.0000 | 1.0000 | 1.0000 |
| | AlterRep | 0.9900 | 0.8443 | 0.9114 | 0.8299 | 1.0000 | 0.9070 |
| | FGSM | 1.0000 | 1.0000 | 1.0000 | 1.0000 | 1.0000 | 1.0000 |
| | PGD | 0.8300 | 1.0000 | 0.9071 | 1.0000 | 1.0000 | 1.0000 |
| filler_gap_pp | HDMI | 0.9710 | 0.4412 | 0.6067 | 0.7404 | 0.6275 | 0.6793 |
| | AlterRep | 0.8095 | 0.7389 | 0.7726 | 0.5100 | 0.4582 | 0.4782 |
| | FGSM | 0.9764 | 0.0501 | 0.0954 | 0.4453 | 0.9951 | 0.6153 |
| | PGD | 0.8480 | 0.2290 | 0.3606 | 0.4561 | 0.9310 | 0.6122 |
| filler_gap_subj | HDMI | 1.0000 | 0.4651 | 0.6349 | 1.0000 | 1.0000 | 1.0000 |
| | AlterRep | 1.0000 | 0.5200 | 0.6842 | 1.0000 | 0.5200 | 0.6842 |
| | FGSM | 0.9200 | 1.0000 | 0.9583 | 0.3435 | 1.0000 | 0.5114 |
| | PGD | 0.9050 | 1.0000 | 0.9501 | 0.1787 | 1.0000 | 0.3033 |
| LGD | HDMI | 0.9412 | 0.8117 | 0.8716 | 0.9341 | 0.8538 | 0.8921 |
| | AlterRep | 0.9490 | 0.6536 | 0.7741 | 0.9951 | 0.3234 | 0.4881 |
| | FGSM | 0.5813 | 0.3337 | 0.4240 | 0.4393 | 0.9959 | 0.6097 |
| | PGD | 0.5402 | 0.4149 | 0.4694 | 0.7124 | 0.532 | 0.6091 |

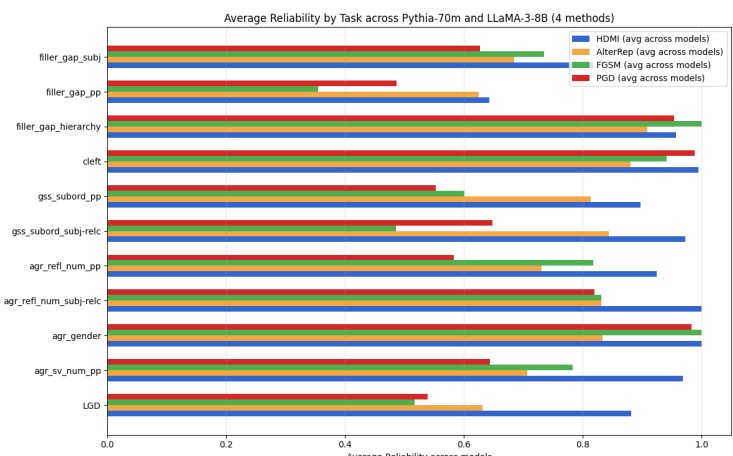

Figure 2: Reliability by tasks. Reliability is averaged across LLaMA and Pythia-70M models.

Following Canby et al. (2024), we report:

**Completeness.** Let $p_c^{\text{after}} = \text{softmax}(probe_{Z_c}(\tilde{h}_\ell(x)))$, where the softmax is taken over $|\mathcal{K}_c|$ classes. The desired distribution after applying the interventional operator $f_\theta(h_\ell(x), z \to z')$ is the one–hot $e_{z'} \in \{0,1\}^{|\mathcal{K}_c|}$, where $[e_{z'}]_k = 1$ if $k = z'$, and zero otherwise; we define

$$\text{Comp} = 1 - d_{\text{TV}}(p_c^{\text{after}}, e_{z'}), \qquad d_{\text{TV}}(p, q) = \tfrac{1}{2}\|p - q\|_1,$$

where $d_{\text{TV}}(p, q)$ is the total variation (TV) distance between distributions $p$ and $q$.

**Selectivity.** We measure selectivity with TV distance between the post- and pre-intervention validation-probe distributions and normalize by the maximum possible TV shift, which is defined as $m(p)$ below. Let $p_e^{\text{before}} = \text{softmax}(probe_{Z_e}(h_\ell(x)))$ and $p_e^{\text{after}} = \text{softmax}(probe_{Z_e}(\tilde{h}_\ell(x)))$. We report the selectivity score

$$\text{Sel} = 1 - \frac{d_{\text{TV}}(p_e^{\text{after}}, p_e^{\text{before}})}{m(p_e^{\text{before}})}, \quad m(p) = \max\{1 - \min_i p_i, \ \max_i p_i\}.$$

**Reliability.** The harmonic mean $\text{Rel} = \frac{2\,\text{Comp}\times\text{Sel}}{\text{Comp}+\text{Sel}}$.

For each split, we hold out a random $20\%$ subset of that split as internal validation to report the probe accuracy. The probe accuracy for our experiments in different tasks is above 90% except for some tasks on the Pythia-70M model, where the accuracy of the interventional $Z_c$ probe (evaluated on its holdout within the interventional split) dropped to approximately 70%, which made Alter-Rep/FGSM/PGD updates unstable. To stabilize these baselines, we adjusted the train/validation ratio within the interventional split, allocating a larger share to probe training and a smaller share to its internal validation, which improved probe accuracy. It is noteworthy to emphasize that HDMI is probe-free and does not rely on an interventional probe at all; consequently, its performance is independent of the size of the interventional split and of interventional probe accuracy.

Table 1 (on LLaMA-3-8B-Instruct and Pythia-70M) reports Completeness, Selectivity, and Reliability across LGD and the CausalGym suites for four methods: HDMI (ours), AlterRep, FGSM, and PGD. More experiments are provided in Appendix A. Figure 2 summarizes the average Reliability across models for all four methods.

In agreement suites (`agr_*`), HDMI is consistently strong. It achieves perfect Completeness and higher Reliability except in `agr_refl_num_pp` suite, where FGSM outperforms in Selectivity. On subordination and clefting, HDMI remains competitive or better in Reliability. On LLaMA's subordination suites, FGSM is often strong and PGD's performance is mixed; both drop substantially on Pythia-70M. HDMI remains consistent across models. Filler–gap remains the most challenging family. On `filler_gap_pp` (LLaMA), AlterRep surpasses HDMI, while FGSM/PGD underperform markedly. In contrast, on Pythia-70M `filler_gap_pp`, HDMI is stronger than AlterRep, and

- Input sequence: Tell me a story about a girl who loves the stars. There once was a girl named Luna who lived in a small town surrounded by a sky full of twinkling stars. From a young age, Luna was fascinated by the stars and would often sneak out of her bedroom window at night to gaze up at the celestial bodies.
- LA-HDMI generated: Tell me a story about the owl who loves the sun. There once was a wise old owl who lived in a small town surrounded by a dense forest. The owl, named Hoot, was, unlike most owls, who are nocturnal, he had a strange love for the sun. Hoot would often be out of his nest, early in the morning, to watch the sun.

- Input sequence: Today, he was upset and left the room.

- LA-HDMI generated: Today, we were upset and left the room.

(a) The input text (top) contains the source words *girl* and *stars*. The user supplies inline edits, striking out the source tokens and inserting ***owl*** and ***sun***. LA-HDMI steers the hidden state of not only these words but also the words before to produce a fluent bottom sentence. Hence the model generates ***the*** instead of ***a*** while keeping the rest of the wording consistent.

(b) The input text (top) contains the source word *was*. The user supplies inline edits with target word ***were***. LA-HDMI performs lookahead steering of the hidden state, hence the model generates ***we*** beforehand.

Figure 3: HDMI editing examples. Left (a) and right (b) show two different edit realizations.

FGSM/PGD performs slightly better. Reliability of HDMI is consistent across models, while others show mixed Reliability and are sensitive to the quality of the *interventional $Z_c$* probe they target.

Taken together, the results indicate that a simple, per instance margin ascent (equation 1) provides a strong performance—high Completeness with competitive Selectivity—translating into higher Reliability on most tasks and across models (Figure 2). The remaining gaps, notably in `filler_gap_pp` on LLaMA, suggest promising directions for the multi-token objective (equation 2), finer layer selection, and step-size scheduling to further improve Selectivity without sacrificing Completeness.

### 6.2 LOOKAHEAD HDMI

In this section, we show text-editing examples with LA-HDMI. Figure 3(a) illustrates LA-HDMI editing with two simultaneous token substitutions (*girl→owl*, *stars→sun*). HDMI applies a next-step, head-aware margin ascent at each decoding step with the expected embedding, allowing gradients to "look ahead" through the softmax–embedding–transition path (Sec. 5). In this example, lookahead successfully steers the model to realize the user's edits while retaining fluency, including the necessary local contextual adjustments (e.g., the preceding article for *owl*) without over-editing the rest of the sentence. Figure 3(b) shows another example where the user asked for change *was→were*. LA-HDMI performs lookahead steering before generating *he* and puts *we* instead, which is grammatically correct.

In general, we observe that this lookahead mechanism is sensitive to hyperparameters such as the horizon $S_{\max}$, temperatures $\beta_g$ and $\beta_f$, step size $\alpha$, and regularization coefficient $\lambda_{\text{fact}}$. When the token that must be adapted is not close to the edited token—for example a determiner, auxiliary, or other agreement carrier several positions away—the product of Jacobians along the expected-embedding path can attenuate the signal, and the gradient may vanish before reaching the earlier position. In such cases, careful fine-tuning (e.g., modestly increasing $S_{\max}$, using a slightly higher $\beta_g$ to reduce peaky distributions, or scheduling $\alpha$) tends to improve edit realization without sacrificing fluency. There are some cases where the lookahead mechanism fails. This is an interesting direction for future work. Please refer to Appendix A for more details.

### 7 CONCLUSION

We introduced HDMI, a probe-free, inference-time causal probing method that modifies hidden states via a simple logit-margin, and LA-HDMI for fluent, lookahead text editing. HDMI attains high completeness and selectivity, yielding strong reliability versus probe-driven baselines. Limitations include the sensitivity of hyperparameters and attenuation of lookahead gradients in text editing. Future work includes adaptive layer selection, step-size scheduling, and broader applications of the multi-token objective.

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

# A MORE EXPERIMENTS

## A.1 MORE CAUSALGYM TASKS

More experiments on causalgym tasks are provided below: On `agr_gender`, all methods are near ceiling on both models, with HDMI matching the best. On `cleft` and `filler_gap_hierarchy`, methods are generally at or near ceiling; HDMI ties or leads in most cases.

Table 2: CausalGym and LGD: Completeness, Selectivity, and Reliability for HDMI (ours), AlterRep, FGSM, and PGD on Meta-Llama-3-8B-Instruct and EleutherAI/pythia-70m. Higher is better.

| Task | Method | Meta-Llama-3-8B-Instruct | | | EleutherAI/pythia-70m | | |
|---|---|---|---|---|---|---|---|
| | | Completeness | Selectivity | Reliability | Completeness | Selectivity | Reliability |
| agr_sv_num_ obj-relc | HDMI | 1.0000 | 1.0000 | 1.0000 | 1.0000 | 1.0000 | 1.0000 |
| | AlterRep | 0.9800 | 1.0000 | 0.9899 | 0.9600 | 1.0000 | 0.9796 |
| | FGSM | 1.0000 | 1.0000 | 1.0000 | 1.0000 | 1.0000 | 1.0000 |
| | PGD | 0.6307 | 1.0000 | 0.7736 | 1.0000 | 1.0000 | 1.0000 |
| agr_gender | HDMI | 1.0000 | 1.0000 | 1.0000 | 1.0000 | 1.0000 | 1.0000 |
| | AlterRep | 1.0000 | 1.0000 | 1.0000 | 0.5000 | 1.0000 | 0.6667 |
| | FGSM | 1.0000 | 1.0000 | 1.0000 | 1.0000 | 1.0000 | 1.0000 |
| | PGD | 0.9350 | 1.0000 | 0.9664 | 1.0000 | 1.0000 | 1.0000 |
| npi_any _subj-relc | HDMI | 1.0000 | 1.0000 | 1.0000 | 1.0000 | 1.0000 | 1.0000 |
| | AlterRep | 1.0000 | 1.0000 | 1.0000 | 0.5000 | 1.0000 | 0.6667 |
| | FGSM | 1.0000 | 1.0000 | 1.0000 | 0.5000 | 1.0000 | 0.6667 |
| | PGD | 0.9600 | 1.0000 | 0.9796 | 0.4999 | 1.0000 | 0.6666 |
| cleft | HDMI | 1.0000 | 0.9800 | 0.9899 | 1.0000 | 1.0000 | 1.0000 |
| | AlterRep | 0.8603 | 0.9800 | 0.9163 | 0.7300 | 1.0000 | 0.8440 |
| | FGSM | 1.0000 | 1.0000 | 1.0000 | 0.7900 | 1.0000 | 0.8827 |
| | PGD | 0.9946 | 1.0000 | 0.9973 | 0.9599 | 1.0000 | 0.9796 |
| filler_gap_ hierarchy | HDMI | 1.0000 | 0.8414 | 0.9138 | 1.0000 | 1.0000 | 1.0000 |
| | AlterRep | 0.9900 | 0.8443 | 0.9114 | 0.8299 | 1.0000 | 0.9070 |
| | FGSM | 1.0000 | 1.0000 | 1.0000 | 1.0000 | 1.0000 | 1.0000 |
| | PGD | 0.8300 | 1.0000 | 0.9071 | 1.0000 | 1.0000 | 1.0000 |
| LGD | HDMI | 0.9412 | 0.8117 | 0.8716 | 0.9341 | 0.8538 | 0.8921 |
| | AlterRep | 0.9490 | 0.6536 | 0.7741 | 0.9951 | 0.3234 | 0.4881 |
| | FGSM | 0.5813 | 0.3337 | 0.4240 | 0.4393 | 0.9959 | 0.6097 |
| | PGD | 0.5402 | 0.4149 | 0.4694 | 0.5402 | 0.4149 | 0.4694 |

## A.2 ABLATION: REMOVING THE MARGIN TERM (TARGET-ONLY OBJECTIVE)

We hypothesize that the margin objective in equation 1 is critical for reliably flipping the targeted property. To test this, we removed the source target term and optimized a *target-only* objective that promotes the logit of the target token but does not explicitly demote the source token:

$$\mathcal{L}_{\text{target-only}}(x) = \phi(x)_\tau \quad \text{vs.} \quad \mathcal{L}_{\text{margin}}(x) = \phi(x)_\tau - \phi(x)_\sigma.$$

We evaluated on the LGD corpus with Meta-Llama-3-8B-Instruct. The results are summarized below and compared against the original HDMI numbers reported in Table 1.

## A.3 CASE STUDY WHEN LOOKAHEAD BACKPROPAGATION FAILS

Consider the following input, edited, and generated sequences.

- *Factual*: *Today, the boy with the red shirt took his own bag and left the room.*

Table 3: LGD (Meta-Llama-3-8B-Instruct) ablation of the HDMI objective. The margin-based HDMI row reproduces the LGD numbers from Table 1.

| Objective | Completeness | Selectivity | Reliability |
|---|---|---|---|
| Target-only ($\phi_\tau$) | 0.7407 | 0.8145 | 0.7758 |
| Margin (equation 1) | 0.9412 | 0.8117 | 0.8716 |
| $\Delta$ (target-only $-$ margin) | $-0.2005$ | $+0.0028$ | $-0.0958$ |

Table 4: Hyperparameters with the range of values used in the tasks.

| Hyperparameter | Range |
|---|---|
| `hdmi_alpha` | 1 |
| `hdmi_inner_steps` | 30 |
| `alterrep_alpha` | $\{0.1, 0.5\}$ |
| `alterrep_inlp_rank_apply` | 32 |
| `probe_epochs` | $\{75, 100\}$ |
| `probe_lr` | $1 \times 10^{-2}$ |
| `probe_weight_decay` | $1 \times 10^{-6}$ |
| `probe_batch_size` | 256 |
| `probe_hidden` | 256 |
| `inlp_epochs` | $\{50, 100\}$ |
| `inlp_lr` | $10^{-2}$ |
| `pgd_steps` | $\{40, 50, 100\}$ |
| `inlp_rank` | 32 |
| `gbi_norm` | $\ell_\infty$ |
| `epsilon` | $\{0.5, 1, 10\}$ |

- *Edited*: *Today, the boy with the red shirt took her own bag and left the room.*

- *Generated*: *Today, the boy with the red shirt took her own bag and left the room. She was very upset and felt not want to talk to anyone.*

This example highlights both the strength and the pitfalls of our lookahead steering. On the positive side, HDMI realizes the intended local edit (*his→her*) with high *completeness*. In the example, the next sentence produces *She*, indicating that the hidden state has been steered to encode the counterfactual gender value; hence, the *targeted* causal property $Z_c$ (here: pronominal gender at the next relevant decoding step) has been changed completely. However, the subject remains *the boy*, hence, look ahead and back propagating the gender change has failed here, and reduced fluency.

In practice, when the earlier word (subject here) that must adapt is not adjacent to the edit (e.g., a preceding determiner or auxiliary several tokens away), the chain of Jacobians along the expected-embedding path can attenuate, and the gradient signal to that earlier position can effectively vanish. We have observed this with articles and agreement carriers near noun edits: unless $S_{\max}$, step size, and inner steps are tuned, the previous tokens may fail to adapt (or the model may overcompensate elsewhere). This is a promising direction for future work.

## B HYPERPARAMETERS

For hyperparameter tuning, we performed a grid search, systematically exploring a predefined range of values for each parameter. In the following tables, we provide the fine-tuned parameters for each task.

Probes are linear or 1–hidden–layer MLPs (hidden size selected from $\{64, 256, 512\}$ by validation accuracy), optimized with AdamW for 100 epochs, weight decay $10^{-6}$, batch size 256. To avoid leakage, the interventional $Z_c$ probe is trained strictly on the interventional split; $probe_{Z_c}/probe_{Z_e}$ are trained strictly on the validation–probe split.

## C  SYSTEM CONFIGURATION

HOST AND OS

- OS: Ubuntu 22.04.4 LTS
- Kernel: Linux 6.8.0-59-generic (x86_64)

COMPUTE

- CPU: AMD EPYC 9454, 2 sockets, 48 cores/socket, 2 threads/core (192 logical)
- Memory: 1.5 TiB RAM
- GPU: NVIDIA H100 (80 GB HBM3)

STORAGE

- Local NVMe aggregate: ∼8.6–8.7 TB (XFS)

## D  USE OF LARGE LANGUAGE MODELS

We used Large Language Models (LLMs) to aid or polish the manuscript text. Specifically, LLMs were used to improve grammar, phrasing, and clarity of exposition; they were also used for code debugging.

