# OpenReview forum: "Inference Time Causal Probing in LLMs"
_ICLR.cc/2026/Conference — Submitted to ICLR 2026_

### Official Review · Reviewer_uVoy · 2025-10-27

**Soundness:** 2
**Presentation:** 2
**Contribution:** 2
**Rating:** 4
**Confidence:** 3

**Summary:**

The author proposed LookAhead Hidden-state Driven Margin Intervention (LA-HDMI), a text-editing approach that steers the model’s hidden states toward the desired modification while preserving overall fluency.

**Strengths:**

1. The proposed implementations, particularly those designed to maintain model prediction fluency and prevent output deviation, are well-motivated and demonstrate certain potential for application in future research if they are originally proposed here.
2. Experimental results show that the proposed method consistently outperforms the baselines by a significant margin. The overall design and the comprehensive ablation study presented in the appendix effectively support this claim.

**Weaknesses:**

1. It is unclear to me whether this work still falls within the scope of probing studies, which typically investigate whether learned representations can be linearly explained. As noted in Footnote 1 (line 50), the authors focus more on text editing rather than probing, and this distinction could be clarified further.
2. In the LookAhead HDMI formulation, the method requires the input and output sequences to have the same number of tokens. This constraint prevents edits that change token length. E.g., transforming apple into shoes here: “This is **an** *apple*” into “This is **a pair of** *shoes*” limits the method’s generalization and fluency flexibility.

**Questions:**

The paper’s writing could be improved for clarity and readability. For instance, it would be better to introduce the symbols W and b earlier (line 225), break down the lengthy paragraph spanning lines 232–247, and highlight the best performance results in the benchmark tables.

---

> ### Author Response · Authors · 2025-11-23
>
> We thank the reviewer for their valuable comments and address them below:
>
> > *It is unclear to me whether this work still falls*
>
>
> Our work sits within causal probing, which we distinguish from correlational probing that focuses on linear decodability. Correlational probing asks whether a property $Z$ can be linearly decoded from $h_\ell$; causal probing asks whether {intervening} on $h_\ell$ to change $Z$, leads to a change in the next-token prediction. HDMI is an inference-time, probe-free causal intervention operator: it uses the model head’s gradient to steer $h_\ell$ so that the target continuation becomes more likely while we measure completeness/selectivity following the "reliable causal probing" protocol [1]. The linear “validation probes” in our experiments are only for readout/measurement of $Z_c$ and $Z_e$, not for designing the intervention. As shown in Table 1, our method achieves consistently higher completeness and selectivity and, in most cases, higher reliability across datasets—all without any additional supervised‑probe training. Moreover, Appendix A.3 includes an ablation demonstrating that our margin‑loss objective is very effective for these results and outperforms the simpler alternatives.
>
> LA-HDMI, the lookahead variant, is presented as an extension of the same intervention mechanism to fluently edit text without fine-tuning the model. To avoid any ambiguity, we will also revise the footnote to prevent the impression that our primary goal is editing; the primary contribution is a probe-free, causal probing method, and LA-HDMI illustrates one practical use case.
>
> > [1] Canby, M., Davies, A., Rastogi, C., & Hockenmaier, J. (2024). How reliable are causal probing interventions?
>
> ------------------------
>
> > *In the LookAhead HDMI formulation, the method requires the input and output sequences to have ....*
>
>
> LA‑HDMI does not enforce the edited continuation to have the same length as the original. As shown in Fig. 3a, the edited and the factual/reference outputs can differ in length. We intervene on hidden states only within the user‑specified edit window (up to step T); beyond T, decoding proceeds normally without intervention until the model’s usual stopping criterion (e.g., EOS/stop rule). We will clarify this behavior in the revised manuscript and update the caption of Fig. 3 accordingly.
>
> -----------------------
>
> > *The paper’s writing could be improved ...*
>
> We thank the reviewer for these concrete suggestions. We already introduced $W_U$ and bias $b$ earlier in Section 3 (``PROBLEM SETTING AND NOTATIONS''), however in the revision we will (i) remind the reader in line 225, (ii) split the long paragraph that introduces our margin objective into smaller, titled subparts (source/target definitions, loss, gradient, and update), and (iii) highlight best results in all benchmark tables (best in \textbf{bold}, second best \underline{underlined}), and state this in the captions.
>
> --------------------
>
> Thank you once again for your feedback. We hope our responses have addressed your concerns and sincerely appreciate your consideration. If there are any additional questions or points that require clarification, please do not hesitate to let us know.

---

> > ### Comment · Reviewer_uVoy · 2025-11-23
> >
> > I thank the authors for some solid rebuttal responses, which resolve most of my concerns. I have raised my evaluations for one level.

---

### Official Review · Reviewer_pf6F · 2025-10-31

**Soundness:** 3
**Presentation:** 3
**Contribution:** 2
**Rating:** 4
**Confidence:** 4

**Summary:**

1. The authors introduce causal probing, a technique that leverages a causal intervention to steer LLMs, using Hidden-state Driven Margin Intervention (HDMI), a probe-free, gradient-based technique that directly steers hidden states using the model’s native output.
2. The authors also introduce a look ahead variant, (LA-HDMI) for text editing that backpropagates through the softmax embeddings, modifying the current hidden state so that the likelihood of user-specified tokens increases in next token generations while preserving fluency.
3. The authors evaluate their method on "completeness" -- i.e elicitation of the new property and "selectivity", i.e. localized editing so that extant properties are not impacted.

**Strengths:**

1. The authors introduce inference time causal probing. Their method, Hidden-state Driven Margin Intervention (HDMI) performs a lightweight gradient-based update to hidden states that shifts the model’s output distribution
2. The multi-token text edit changes are very interesting, and could be a useful direction in the current interpretability literature.

**Weaknesses:**

1. My primary and main concern is that the authors' core contribution -- HDMI -- seems to be a reinvention of activation patching[1]/it's gradient based approximation, attribution patching[2] (!) -- it uses using gradient ascent to shift logits from one output type to another instead of direct patching. I think reframing the paper as a comparative study of probing vs patching based approaches, and claiming the advantages of causal vs correlational interventions would be acceptable, but claiming the invention of a new method seems overstated. Could the authors clarify how HDMI is different from activation/attribution patching[1,2], FutureLens[6] as well as representation fine-tuning[5]? In attribution patching, a gradient ascent/descent can be applied to the activations of the attention heads in order to identify important components in the model.
2. The second issue is that given the oversight in [1], the authors have overlooked several important papers/baselines that do use causal interventions [3], [4] and others -- there's an entire subfield using these patching methods. Particularly, [4] tackles the very same problem of correlational and causal probing, albeit in a different setting.
3. How were the reliability and selectivity metrics obtained? Have the authors investigated the effect of an HDMI edit on existing model capabilities like MMLU or GSM-8k?

**Questions:**

1. My primary and main concern is that the authors' core contribution -- HDMI -- seems to be a reinvention of activation patching[1]/it's gradient based approximation, attribution patching[2] (!) -- it uses using gradient ascent to shift logits from one output type to another instead of direct patching. I think reframing the paper as a comparative study of probing vs patching based approaches, and claiming the advantages of causal vs correlational interventions would be acceptable, but claiming the invention of a new method seems overstated. Could the authors clarify how HDMI is different from activation/attribution patching[1,2], FutureLens[6], as well as representation fine-tuning[5]? In attribution patching, a gradient ascent/descent can be applied to the activations of the attention heads in order to identify important components in the model.

2. The second issue is that given the oversight in [1], the authors have overlooked several important papers/baselines that do use causal interventions [3], [4] and others -- there's an entire subfield using these patching methods. Particularly, [4] tackles the very same problem of correlational and causal probing, albeit in a different setting.

I will increase my score if I am convinced by the authors' rebuttal to 1 and 2 above.

4. Could the authors expand what LGD is and cite it at the very first mention?
5. Could the authors add a limitations section on potential improvements to HDMI?
6. It is unclear how completeness is measured in the multi-token setting, since the single-token setting uses a one-hot encoding based evaluation. How would this be determined in the multi-token settings?. Could authors clarify the same?
7. Is there a significant improvement, based on Table 1? Could the authors bold the top values to improve readability?
8. How does HDMI affect existing capabilities, such as on MMLU/GSM-8k etc, i.e on tasks unrelated to the task settings the intervention was applied for?

[1] Jesse Vig, Sebastian Gehrmann, Yonatan Belinkov, Sharon Qian, Daniel Nevo, Yaron Singer, and Stuart Shieber. Investigating gender bias in language models using causal mediation analysis. In Advances in Neural Information Processing Systems (NeurIPS), 2020.

[2] Syed, Aaquib, Can Rager, and Arthur Conmy. "Attribution patching outperforms automated circuit discovery, 2023." URL https://arxiv. org/abs/2310.10348.

[3] Meng, Kevin, et al. "Locating and editing factual associations in gpt." Advances in neural information processing systems 35 (2022): 17359-17372.

[4] Huang, Jing, et al. "Internal Causal Mechanisms Robustly Predict Language Model Out-of-Distribution Behaviors." arXiv preprint arXiv:2505.11770 (2025).

[5] - Wu, Zhengxuan, et al. "Reft: Representation finetuning for language models." Advances in Neural Information Processing Systems 37 (2024): 63908-63962.

[6] Pal, Koyena, et al. “Future Lens: Anticipating Subsequent Tokens from a Single Hidden State.” Proceedings of the 27th Conference on Computational Natural Language Learning (CoNLL), Association for Computational Linguistics, 2023, pp. 548–60. Crossref, https://doi.org/10.18653/v1/2023.conll-1.37.

---

> ### Author Response · Authors · 2025-11-23
>
> We thank the reviewer for their valuable comments and address them below:
>
> > *My primary and main concern is that the authors' core contribution -- HDMI -- seems to be a reinvention of...ould the authors clarify how HDMI is different from activation/attribution patching[1,2], FutureLens[6] as well as representation fine-tuning[5]? ...The second issue is that given the oversight in [1], the authors have overlooked several important papers/baselines that do use causal interventions [3], [4] *
>
>
>
>
> Although the works mentioned by the reviewer are related to our topic, each belongs to a distinct line of literature and differs in key ways from our approach; none addresses the same problem we study.
>
>
> To compare the mentioned papers, we use the following terminology: An activation is the intermediate data a model produces while processing an input. In a transformer, this includes the hidden state at each layer and token, plus the outputs of attention heads and MLPs.
> Steering means making small changes to a model’s behavior by adjusting its activations, without changing its weights. “Control” means making a language model generate text that satisfies a user-specified attribute and avoids an undesired attribute. Examples include positive vs. negative sentiment, non-toxic vs. toxic content, and target topics (e.g., Sports, World). Therefore, steering is one way to achieve ``Control".
> Causal-probing methods intervene on hidden states to change linguistic properties; therefore, steering the hidden state is a tool to make the intervention. Not every steering method has the same objective as causal probing. Below, we explain how each paper differs from our approach.
>
>
> ## Relation with mediation analysis and activation patching.
>
>
> ROME (Meng et al.,2022) edits model parameters (rank-one weight updates in mid-layer MLPs) to permanently change specific factual associations, validated for generalization and specificity. It is a model-editing method, not an inference-time causal probing method.
>
>
>
> Causal mediation analysis aims to measure how a treatment effect is mediated by intermediate variables. (Vig et al., 2020) uses causal mediation analysis to quantify how specific mediators (neurons/heads) transmit an effect (e.g., gender bias) from input to output; they measure the direct and indirect effects and analyze mechanisms. They do not steer/control text at runtime.
> What we do is per-instance inference-time interventions on hidden states to directly change the next-token distribution at a chosen position.
>
> Activation patching refers to replacing the activations from one model's forward pass with the activations from a different forward pass. A diagnostic technique where the model is run on two inputs—one “clean” (where it gets the task right) and one “corrupted” (where it doesn’t)—and then a specific activation is copied from the clean run into the corrupted run. If the model’s output improves, that activation (and the component producing it) was causally important.
> (Syed et al., 2023) runs a “clean” and a “corrupted” forward pass, and “patches” selected activations from the clean into the corrupted run to measure restoration of behavior, thereby identifying mediators (e.g., specific heads/ MLPs/ positions).
> Attribution patching (Syed et al., 2023) is a scalable, linearized approximation to activation patching that assigns attribution scores to all edges in a transformer’s computational graph; it uses two forward passes and one backward pass, and is designed to recover circuits (which is the small set of model components and connections that causes a specific output/behavior) efficiently, not to steer a particular hidden state at inference time like HDMI. The gradients are used to score each edge by combining that gradient with the difference between “clean” and “corrupted” activations. Therefore, the task is completely different than causal probing.
>
> ## Relation to Representation fine‑tuning (ReFT).
>
> ReFT (Wu et al., 2024) is training time representation fine‑tuning
> that edits hidden representations at chosen layers/ positions. It updates parameters (or small trainable modules) to change the model’s behavior persistently. So it solves a different problem (persistent model changes for knowledge editing).

---

> ### Author Response · Authors · 2025-11-23
>
> ## Relation to FutureLens.
>
> FutureLens (Pal et al., 2023) is a descriptive method: it asks what future tokens can already be read from a single hidden state, showing that one state can predict tokens several steps ahead. It focuses on describing what is present in the hidden representation, not performing gradient-based adjustments to the hidden state. It does not have contrastive loss like us, and solves a different problem.
> By contrast, our LA-HDMI uses the differentiable “expected-embedding” path to actively intervene on hidden states toward user‑specified future edit with a margin loss. We define a margin objective that raises the score of the desired future tokens and lowers the original ones, and we take gradient steps on hidden states during decoding. To keep the text fluent, we generate with a low temperature while computing gradients with a higher temperature so that useful signal still flows back to earlier states. In the revised version, we will discuss the differences between causal probing (the problem considered in our paper) and other problems addressed in the literature.
>
> ----------------
>
> > *How were the reliability and selectivity metrics obtained?*
>
> In section 6.1, we thoroughly explain how we compute reliability and selectivity. As mentioned in the paper, our motivation comes from (Canby et al. 2024). They primarily answer
> Does the intervention push the target properties $Z_c$ toward the desired value? (Completeness)
> Does it avoid altering unrelated properties $Z_e$ of the context?(Selectivity)
> We did not evaluate MMLU or GSM-8K in this work. MMLU and GSM-8K are potentially applicable to HDMI if we annotate the data, and for each item, specify the causal property and source/target token sets. Building these annotations is a promising future direction to strengthen our evaluation.
>
> ---------------
>
> > **Could the authors expand what LGD is and cite it at the very first mention?**
>
> We used the subject-verb agreement dataset of [7] (often abbreviated {LGD} after Linzen--Goldberg--Dupoux), which provides English sentences where the next-token verb form depends on the grammatical number of the subject.
> >[7] Linzen, T., Dupoux, E., & Goldberg, Y. (2016). Assessing the ability of LSTMs to learn syntax-sensitive dependencies.
> ---------------
>
> > *Could the authors add a limitations section...*
>
> We will add a Limitations section that covers: (i) hyperparameter sensitivity:
>     Reliability depends on the step size $\alpha$, number of inner steps $K$, the lookahead horizon $S_{\max}$, and the temperatures used during decoding ($\beta_f$ for the forward pass and $\beta_g$ for the lookahead gradient path). Simple schedules (e.g., decaying $\alpha$, adaptive $\beta_f$ and $\beta_l$) can improve stability and fluency; it can adjust the trade-off between completeness and selectivity.
> (ii) Gradient attenuation in LA‑HDMI. In the lookahead path that uses expected embeddings, the product of Jacobians can attenuate gradients to earlier tokens. Careful choices of $S_{\max}$ and temperature parameters would help. We were thinking that choosing adaptive temperature parameters and step size until a margin threshold is reached may further mitigate attenuation; we leave a systematic study to future work.
>
> -------------------

---

> > ### Author Response · Authors · 2025-11-23
> >
> > > *It is unclear how completeness is measured in the multi-token setting...*
> >
> > Completeness is defined in the latent property space and is unchanged in the multi-token setting. We read out the post-intervention distribution over $Z_c$ with the validation probe and compare it to the one-hot of the target class. For a binary property (e.g., number: singular vs.\ plural), this reduces to the post-intervention probability of the target class; for multi-class, we use $1 - d_{\mathrm{TV}}(p_c^{\text{after}}, e_{z'})$. Although a property value (e.g., ``plural'') may be realized by multiple tokens in the multi-token objective, the property label remains a single class, so the Completeness definition is the same. Both single- and multi-token objectives flip the same underlying feature value (e.g., singular $\rightarrow$ plural).
> >
> >
> > -----------------
> >
> > >*Is there a significant improvement...*
> >
> > As shown in Table 1, our method achieves consistently higher completeness and selectivity and, in most cases, higher reliability across datasets—all without any additional supervised‑probe training. Moreover, Appendix A.3 includes an ablation demonstrating that our margin‑loss objective is very effective for this results and outperforms the simpler alternatives.
> >     We will bold the best values and underline second-best to improve readability. We will also report the number of suites where HDMI attains the best Reliability
> >
> > -------------------
> > > *How does HDMI affect existing capabilities...*
> >
> >
> > HDMI does not modify model parameters and is only applied locally at inference time (specific layer/position) when an intervention is considered; therefore, we do not expect degradation on unrelated tasks unless one deliberately applies HDMI during those tasks.
> >
> > -------------
> >
> > Thank you once again for your feedback. We hope our responses have addressed your concerns and sincerely appreciate your consideration. If there are any additional questions or points that require clarification, please do not hesitate to let us know.

---

> ### Comment · Reviewer_pf6F · 2025-11-25
>
> I was concerned that the authors were unaware of the literature at the time of my initial review. I am only more convinced that they don't understand the existing literature from their response to my review, which leaves me deeply concerned about the quality and novelty of this work. Further, instead of highlighting the differences between their work and the existing literature, they describe what the existing literature is. This doesn't help their case.
>
> > Causal mediation analysis aims to measure how a treatment effect is mediated by intermediate variables. (Vig et al., 2020) uses causal mediation analysis to quantify how specific mediators (neurons/heads) transmit an effect (e.g., gender bias) from input to output; they measure the direct and indirect effects and analyze mechanisms. They do not steer/control text at runtime. What we do is per-instance inference-time interventions on hidden states to directly change the next-token distribution at a chosen position.
>
> This is false. Here is a non-exhaustive set of papers from just the last 12 months that use causal mediation analysis for steering/control:
> - Sharma, Arnab Sen, et al. "LLMs Process Lists With General Filter Heads." arXiv preprint arXiv:2510.26784 (2025).
> - Zhao, Jiachen, et al. "Llms encode harmfulness and refusal separately, 2025." URL https://arxiv. org/abs/2507.11878.
> - Feucht, Sheridan, et al. "The Dual-Route Model of Induction." arXiv preprint arXiv:2504.03022 (2025).
> - Sankaranarayanan, Aruna, Dylan Hadfield-Menell, and Aaron Mueller. "Disjoint processing mechanisms of hierarchical and linear grammars in large language models." arXiv preprint arXiv:2501.08618 (2025).
> - Ahsan, Hiba, et al. "Elucidating Mechanisms of Demographic Bias in LLMs for Healthcare." arXiv preprint arXiv:2502.13319 (2025).
> - Prakash, Nikhil, et al. "Fine-tuning enhances existing mechanisms: A case study on entity tracking, 2024." URL https://arxiv. org/abs/2402.14811.
>
> > Attribution patching (Syed et al., 2023) is a scalable, linearized approximation to activation patching that assigns attribution scores to all edges in a transformer’s computational graph; it uses two forward passes and one backward pass, and is designed to recover circuits (which is the small set of model components and connections that causes a specific output/behavior) efficiently, not to steer a particular hidden state at inference time like HDMI. The gradients are used to score each edge by combining that gradient with the difference between “clean” and “corrupted” activations. Therefore, the task is completely different than causal probing.
>
> Not true. Attribution patching, given it's speed can do this ranking/steering process at run-time.
>
> Further all methods allow for intervening on activations at inference-time, a property I have personally used extensively in my own research. The authors need to develop a more rigorous familiarity with the existing literature. Not only are they unaware about the existing literature, they seem to have a misinformed view of the existing literature, which is very concerning.
>
> > In section 6.1, we thoroughly explain how we compute reliability and selectivity. As mentioned in the paper, our motivation comes from (Canby et al. 2024). They primarily answer Does the intervention push the target properties $Z_c$ toward the desired value? (Completeness) Does it avoid altering unrelated properties $Z_e$ of the context?(Selectivity) We did not evaluate MMLU or GSM-8K in this work. MMLU and GSM-8K are potentially applicable to HDMI if we annotate the data, and for each item, specify the causal property and source/target token sets. Building these annotations is a promising future direction to strengthen our evaluation.
>
> This sounds like a non-trivial limitation of your work, since nearly all steering papers today conduct these evaluations to show selectivity. Examples from some seminal works:
>
> - Arditi, Andy, et al. "Refusal in language models is mediated by a single direction." Advances in Neural Information Processing Systems 37 (2024): 136037-136083.
> - Panickssery, Nina, et al. "Steering llama 2 via contrastive activation addition, 2024." URL https://arxiv. org/abs/2312.06681 3.
> - Turner, Alexander Matt, et al. "Activation addition: Steering language models without optimization." arXiv e-prints (2023): arXiv-2308. (See Sec. 4.5)
>
> I am reducing my current score.

---

> > ### Author Response · Authors · 2025-11-26
> >
> > Dear Reviewer,
> >
> > Thank you for the comments. In our response, we were referring to the definition of causal mediation analysis (CMA) in [1, Section 3.2] and the definition of activation patching in [2, Section 3.1]. Since your comment cited these two works, our discussion was scoped to them.
> >
> > Let us clarify what we wrote about [1]. Your statement differs from what we wrote. We stated that mediation analysis aims to measure the effect; [1] also uses interventions (editing the input or patching the internal representation) to measure causal effects cleanly and to pinpoint which internal components actually cause a behavior. Thus, it is more about using interventions to perform mediation analysis than using mediation analysis to intervene. Our work is the converse: we optimize a logit-margin, which is analogous to an “effect” (but is not the same as the definition of effects in [1]), to intervene on hidden states for causal probing using gradient descent, and we extend this to do local edits in multiple decoding times using the softmax embedding in LA-HDMI.
> >
> > Our core contribution is to intervene on the hidden state to change the linguistic properties encoded in that hidden state (locally, not steering the model behavior in general), and we evaluate it not with the output sequence of the model, but only with validation probes (section 6 of our paper) on the modified hidden state; we do not perform two forward passes and patch one hidden state into the other to steer the model output. We also do not permanently train/change model parameters as in ROME [3]. Moreover, that line of work is not categorized as causal probing. Many of the papers you mention leverage causal techniques, but are not causal probing; instead, [8] discusses methods such as AlterRep, INLP, PGD, and FGSM. Many methods in the causal probing literature use a trained probe to intervene on the hidden state to change linguistic features, but with our margin loss we argue that we achieve better completeness and selectivity. Our approach involves automated interventions using a gradient.
> >
> > Moreover, completeness and selectivity were first defined in [8] for causal probing interventions. It cites prior work that uses minimal-pair-based interchange/patching setups, including Vig et al. (2020). It notes that this line of work relies on constructing counterfactual pairs (e.g., two forward passes by swapping gendered words he↔she while holding everything else fixed) to cleanly measure the causal effect of a specific property on model behavior.
> >
> > We will revise the manuscript to make these distinctions explicit.
> >
> > >[8] Canby, M., Davies, A., Rastogi, C., & Hockenmaier, J. (2024). How reliable are causal probing interventions?

---

> > > ### Comment · Reviewer_pf6F · 2025-11-26
> > >
> > > > Let us clarify what we wrote about [1]. Your statement differs from what we wrote. We stated that mediation analysis aims to measure the effect; [1] also uses interventions (editing the input or patching the internal representation) to measure causal effects cleanly and to pinpoint which internal components actually cause a behavior. Thus, it is more about using interventions to perform mediation analysis than using mediation analysis to intervene.
> > >
> > > Thank you for clarifying. This definitely helps. Could you elaborate how your process uses mediation analysis to intervene?
> > >
> > > > Our work is the converse: we optimize a logit-margin, which is analogous to an “effect” (but is not the same as the definition of effects in [1])
> > >
> > > Could you elaborate why the logit-margin you optimize is different from the logit difference attribution used to quantify the indirect effect of a model component in classic CMA?
> > >
> > > > Moreover, completeness and selectivity were first defined in [8] for causal probing interventions. It cites prior work that uses minimal-pair-based interchange/patching setups, including Vig et al. (2020). It notes that this line of work relies on constructing counterfactual pairs (e.g., two forward passes by swapping gendered words he↔she while holding everything else fixed) to cleanly measure the causal effect of a specific property on model behavior.
> > >
> > > Thanks for the clarification. I believe the definitions of selectivity in [8] as well as MMLU/GSM-8k evals suggested are complimentary.

---

> > > > ### Comment · Reviewer_pf6F · 2025-11-26
> > > >
> > > > From the paper,
> > > > > Causal probing addresses the aforementioned limitation by interventions on the hidden state and
> > > > tests whether such perturbations alter the next-token distribution of the model. For instance, if we
> > > > modify the hidden state so that the subject property changes from singular to plural, we would like
> > > > to see whether the model accordingly modifies the verb from singular to plural form. Therefore,
> > > > causal probing investigates not only what information is encoded, but also how it is used in prediction
> > > > (Elazar et al., 2021; Ravfogel et al., 2020; Kumar et al., 2022). 1
> > > >
> > > > Could the authors clarify why this is not the same as causal mediation analysis? CMA also does interventions and measures whether there is a change in the next-token distribution. How is causal probing different? From your responses, it sound like the primary difference is the objective. While CMA aims to find the best intervention sites, causal probing aims to change the output, but methodologically they are identical, i.e. both CMA and causal probing apply intevrentions and measure the change in the output. Is this understanding correct?

---

> > > > > ### Author Response · Authors · 2025-11-26
> > > > >
> > > > > Thank you for the comment. To clarify, let us review the definitions. For a given prompt, [1] defines the "bias" as the probability ratio between the anti-stereotypical and stereotypical tokens (which correspond to the source and target tokens in our margin loss):
> > > > > $$
> > > > > y(\text{prompt}) \;=\; \frac{p(\text{anti-stereotypical}\mid \text{prompt})}{p(\text{stereotypical}\mid \text{prompt})}.
> > > > > $$
> > > > >
> > > > > Natural Indirect Effect (NIE) is the expectation of \(y'/y\):
> > > > > $$
> > > > > \mathrm{NIE} \;=\; \mathbb{E}\left[\frac{y'(\text{prompt})}{y(\text{prompt})}-1\right],
> > > > > $$
> > > > > where $y'(\text{prompt})$ is computed on the same prompt but with the mediator (e.g., a neuron/hidden state) replaced by the value obtained under a different prompt. Thus, \(y'/y\) compares two runs/worlds.
> > > > >
> > > > > In contrast, the margin loss is not a between-runs quantity; it is a single-run quantity. We optimize the log probability ratio (logit difference):
> > > > > $$
> > > > > \log y(\text{prompt}) \;=\; \log p(\text{anti-stereotypical}\mid \text{prompt}) \;-\; \log p(\text{stereotypical}\mid \text{prompt}),
> > > > > $$
> > > > > computed within one run/world.
> > > > >
> > > > > There are different variants of "effects" defined in the CMA literature, and as a statistic logit difference apeared in them. For example, in the above effect definition, the logit difference is actually the log of the probability ratio; in [2], they used logit difference (or loss) as a statistic to capture the change under activation patching and then they approximate it for attribution scoring.
> > > > > In [2] the actual “intervention” is a replacement e_clean → e_corr (interchange-style interventions, activation patching), and gradients are used for scoring and estimating the metric change under activation patching (to quantify mediation). While in our paper gradients are used to intervene/update the hidden states.
> > > > > We can say that the logit difference is the algebraic building block used for different purposes. Up to our knowledge, there is no method in causal probing that updates the hidden state by gradient ascent on the margin objective at inference time.
> > > > >
> > > > > Maybe our sentence ("alter the next-token distribution of the model") in the paper was misleading and we will revise it, but the goal in causal probing is to alter the properties encoded in the hidden representation by intervening on it (property-level interventions), and the main way we realize that the property has been altered and the unrelated properties were not is by using a probe (a classifier that takes the hidden state and returns the property encoded) independent of the next-token distribution, while altering a property can be realized by that too.

---

> ### Comment · Reviewer_pf6F · 2025-11-27
>
> > In contrast, the margin loss is not a between-runs quantity; it is a single-run quantity. We optimize the log probability ratio (logit difference): $$ \log y(\text{prompt}) ;=; \log p(\text{anti-stereotypical}\mid \text{prompt}) ;-; \log p(\text{stereotypical}\mid \text{prompt}), $$ computed within one run/world.
>
> The commonly used approach in CMA-based interpretability right now is the single-run margin loss. For example, when the model is processing "Paris is a city in", one could intervene at a model component on the token "Paris" or the token "in" with the counterfactual token from "Rome is a city in", and then measure $\log p(\text{Italy} | (\text{Paris is a city in})^{intervened})$ - $\log p(\text{France} | (\text{Paris is a city in})^{intervened})$ -- note that this is also a single run quantity.
>
> Good luck with the revisions -- I believe that the main bottleneck is the lack of clarification between causal probing and CMA as well as the differences between your algorithm and CMA-variants.

---

> > ### Author Response · Authors · 2025-11-27
> >
> > Dear Reviewer,
> >
> > Thank you for your comment. We have made a concerted effort to clarify the differences between our work and papers [1] and [2], and we are dedicated to clarifying more if needed. May we kindly ask if the distinction regarding the role of "logit difference" in attribution patching in [2] versus our method is now clear? How about the distinction between our method and that of [3]?

---

### Official Review · Reviewer_G5nM · 2025-10-31

**Soundness:** 4
**Presentation:** 4
**Contribution:** 1
**Rating:** 2
**Confidence:** 4

**Summary:**

This paper presents a method called HDMI that steers a language model generation towards a target text and away from a source text. The experiments are on Causal Gym (a benchmark for localizing syntactic features to directions in activation space) and another corpus based on singular plural syntax agreement. They compare HDMI against some baselines, and HDMI performs best.

Overall, I would say that this paper is very well written and presents all the ideas clearly. However, there are major problems when it comes to contextualizing the work within the current literature and a consequent poor coverage of existing baselines.

This paper demonstrates poor awareness of the following literatures:
- Causal mediation and abstraction analysis of LLMs. For example, https://arxiv.org/abs/2004.12265 and https://arxiv.org/abs/2004.12265; see https://arxiv.org/abs/2301.04709 or https://arxiv.org/pdf/2408.01416 for surveys with citations. Causal gym is an eval designed for these methods, but you use it more like a steering evaluation than a localization evaluation.
- Causal probing is actually already used as a term from a couple other papers https://aclanthology.org/2023.tacl-1.23.pdf and https://arxiv.org/abs/2307.15054
- Steering literature. You cite Turner et al. 2023, but there really is a huge amount of steering literature now and plenty of different methods to steer with. For example, https://arxiv.org/abs/2205.05124, or https://arxiv.org/pdf/2310.06824 or https://arxiv.org/abs/2306.0334. You should compare against difference in means steering!
- Representation fine-tuning: https://arxiv.org/abs/2404.03592. This is a very very similar idea to your paper and needs to be included as a baseline. It's not clear to me how your method exactly differs, and you would need to spell that out.
- Axbench https://arxiv.org/abs/2501.17148 evaluates a bunch of methods for control, and you could evaluate your method on this benchmark to get standardized comparisons against a lot of different methods

Broadly, this seems like a method for steering or editing and needs to be properly evaluated against the methods I point out above. I'm not sure how it will hold up against difference-in-means steering or representation fine-tuning. I think this would require too much experimentation and rewriting for this draft, so I think you should plan to revise and submit elsewhere.

**Strengths:**

See summary

**Weaknesses:**

See summary

**Questions:**

See summary

---

> ### Author Response · Authors · 2025-11-23
>
> We thank the reviewer for their valuable comments and address them below:
>
> Although the topics mentioned by the reviewer are related to our work, there are several key differences.
>
> To compare the papers mentioned by the reviewer, let us first introduce the following terminology: An activation is the intermediate data, a model produces while processing an input. In a transformer, this includes the hidden state at each layer and token, as well as the outputs of attention heads and MLPs.
> Steering refers to making small changes to a model’s behavior by adjusting its activations, without changing the weights of the model. “Control” refers to making a language model generate text that satisfies a user-specified attribute and avoids an undesired attribute. Examples include positive vs. negative sentiment, non-toxic vs. toxic content, and target topics (e.g., Sports, World). Therefore, steering is one way to achieve ``Control".
> Causal-probing methods intervene on hidden states to change linguistic properties; therefore, steering the hidden state is a tool to make the intervention. Not every steering method has the same objective as causal probing.  Below, we explain how each paper differs from our approach.
>
> A) Causal mediation and abstraction analysis
>
> Causal mediation analysis aims to measure how a treatment effect is mediated by intermediate variables. (Vig et al., 2020) uses causal mediation analysis to quantify how specific mediators (neurons/heads) transmit an effect (e.g., gender bias) from input to output; they measure the direct and indirect effects of the mediator. They do not intervene on the hidden states or steer/control text at runtime.
> Activation patching overwrites specific hidden states (often from a so-called “clean” forward pass) and is used to locate which layers/heads/features carry a piece of information. (Geiger et al., 2023–2025) provides a formal theory of causal abstraction and mechanism transformation, unifying methods like activation/path patching, etc.; it is a theoretical framework, not a concrete causal probing/steering algorithm.
>
> What we do is per-instance inference-time interventions on hidden states to directly change the next-token distribution at a chosen position. We are not identifying mediators or decomposing effects.
>
>
> CausalGym is designed to evaluate localization/mediation methods, but it is applicable to be used for our setting as well. Since it provides so-called minimal pairs that can be used as our target and source tokens to perform hidden-state intervention and see if it flips the intended linguistic property while preserving others. The dataset is similar to LGD dataset, for evaluating “intervention”.

---

> > ### Author Response · Authors · 2025-11-23
> >
> > B) Steering literature
> >
> > (Subramani et al., 2022) learns one fixed vector per whole sentence and adds it to all the middle layers so the model can reproduce the target sentence almost exactly; Our goal, however, is causal probing, which is fundamentally different from this task. Moreover, we do not learn a fix reusable vector to construct a sentence; instead, at inference, we change the current hidden state to raise the logit score of a desired next token and lower the original one, flipping a specific property (e.g., $\text{singular}\rightarrow\text{plural}$ or $\text{he}\rightarrow\text{she}$).
> >
> > "n-ADC" paper (, ZILONG HE 2024) is not relevant to language models. It is a work on quadratic lattices (n‑ADC) in algebraic number fields. Therefore, it does not meaningfully compare to causal probing in LLMs.
> > “The Geometry of Truth” (COLM 2024) manipulates hidden states but with different aims and tools: Geometry of Truth asks whether “truth” is linearly encoded. “truth” means the factuality of simple, declarative statements—i.e., whether a statement is true or false in the real world (e.g., “Eighty‑one is larger than fifty‑four” is true; “The city of Denver is in Vietnam” is false).
> > They train simple probes (especially a difference‑in‑means “mass‑mean” probe) on true/false statements and show a linear “truth” direction that can causally flip the model’s answer in certain places in the network. So they have a different objective by learning a dataset-level linear separator for truth vs. falsehood.
> > HDMI is different; without training a probe, we do per‑instance, inference‑time intervention.
> >
> > In general, at a high level, the steering methods typically construct a global “concept/behavior” steering vector (often via contrastive/difference-in-means on hidden states, sometimes combined with safety/robustness protocols), then adds them at chosen layers. Some of them focus on limiting the side effects of steering or learning classifiers to toggle steering. Rather than a single static concept change vector, HDMI computes a per-instance, per-token intervention directly from the model’s output head margin for the exact alternative continuation that is specified; no classifier/probe or dataset of positives/negatives is needed; and LA‑HDMI’s lookahead gradients are designed to preserve forward fluency while achieving multi-token local edits.
> > We evaluate our intervention with completeness (did the target change?) and selectivity (did other things stay the same?) on LGD and CausalGym.
> >
> >
> > C)
> > Relation to ROME/representation fine‑tuning (ReFT)
> >
> > ROME (Meng et al., 2022) edits model parameters (rank‑one weight updates in mid‑layer MLPs) to permanently change specific factual associations, validated for generalization and specificity. It is a model‑editing method, not an inference‑time steering operator.
> > ReFT (Wu et al., 2024) is training‑time representation fine‑tuning to shape hidden spaces with supervision/objectives across data distributions; it updates parameters (or small modules) to achieve desired behaviors.
> > While in HDMI, we have no parameter updates, no retraining, no persistent model edits.
> > Thus, ROME/ReFT solve a different problem (persistent model changes, knowledge editing/learning), whereas HDMI solves a per-step (local), training‑free causal probing.
> >
> > ----------------
> > Thank you once again for your feedback. We hope our responses have addressed your concerns and sincerely appreciate your consideration. If there are any additional questions or points that require clarification, please do not hesitate to let us know.

---

### Official Review · Reviewer_NBLB · 2025-11-06

**Soundness:** 2
**Presentation:** 2
**Contribution:** 2
**Rating:** 4
**Confidence:** 3

**Summary:**

The paper defines a method for intervening on the hidden states of an LLM in order to generate a set of user specified edits. They do this by optimizing over the hidden state to maximize the margin between the original edit and the specified edit. They introduce a variant of the method that can intervene on several hidden states throughout a generation in order to preserve fluency of the decoded text. They evaluate their approach in LGD agreement and CausalGym and show that interventions alter the target properties without modifying unrelated properties.

**Strengths:**

The paper looks at a problem of interest and contributes a novel method. The idea of maximizing a logit margin seems valuable and simple. The single token method seems to reliably induce the desired change in the model's generation (although there is some discrepancy across the different benchmarks). In general, this seems like a valuable and useful contribution to the question of causal interventions on language model hidden states.

**Weaknesses:**

# Presentation

The paper is fairly dense and seems to introduce a lot of notation and modeling that I don't understand the purpose of. E.g., the discussion of the difference between nuisance and causal variables is nice, but it doesn't add much to the description of the method. Ultimately, the central idea of the method (maximize the margin between an original token and a desired token with gradient descent on a hidden state) is quite simple and the paper could be much more direct about explaining the idea.

# Analysis of results

While the paper presents fairly comprehensive results, there is not very much analysis of how to interpret the evaluation numbers. E.g., Table 1 presents a substantial amount of data but the discussion in the text is fairly limited. I think the paper could be improved with an analysis that explains the significance of the results and potentially supports the numerical results with example successes or failures. Reading this, it's not quite clear why these metrics are appropriate or what they truly indicate. From what I can tell, the metrics are several related metrics that determine whether the new hidden state is reliably classified by a probe as an instance of the target class. I think this could be made more concrete and clear to readers.

# Multi-token evaluation

While I find the multi-token lookahead method interesting, the evaluation of it is quite limited. There's no real quantitative evaluation and the primary evidence of success is a potentially cherry-picked example in Fig 3a. I would expect an evaluation of this method to include numbers like: how often the method induces the desired token change, how many tokens are changed, and the fluency/quality of the changed output (evaluated by an LLM-as-a-judge or something similar).

# Significance

My primary concern about the paper is that the authors do not make a clear case for the significance of the method. It is interesting that they can intervene on hidden states, but it's not clear what the practical purpose of this is. Is the goal to steer the model? Is it to understand generation mechanisms? Is there something else? For example, the Arora '24 paper uses CausalGym to identify properties of several different generation properties and show that it emerges in discrete stages. Even if the authors do not want to directly perform analysis like this in their paper, they should more clearly motivate what HDMI can be used for and argue that their method with advance those goals.

**Questions:**

1. **Clarity of motivation:** What is the intended downstream use of HDMI—mechanistic interpretability, controllable generation, or general text editing—and how does the method concretely advance that goal beyond prior activation-steering work?

2. **Interpretation of metrics:** How should readers interpret the reported completeness, selectivity, and reliability scores in practical or conceptual terms? Do these metrics correspond to observable differences in model behaviour or only to probe classification outcomes?

3. **Evaluation of lookahead editing:** Beyond the qualitative examples in Figure 3, can the authors quantify how often LA-HDMI successfully performs the intended edits while maintaining fluency, and how sensitive this success rate is to hyperparameters such as (S_{\max}) and temperature?

4. **Presentation and framing:** Given that the method mainly performs gradient ascent on a logit margin, what is gained by the causal/nuisance variable formalism and extensive notation? Could the same idea be communicated more directly without that framing?

---

> ### Author Response · Authors · 2025-11-23
>
> We thank the reviewer for their valuable comments and address them below:
>
> >*Presentations: The paper is fairly dense...*
>
> We agree and will state the core idea up front with minimal notation. In the revised version, in Section 1, we will be more direct that HDMI applies a simple, gradient step to the hidden state at position $t$ to maximize the logit margin for a user-specified target token $y^*$ over the original $y$. LA-HDMI extends this by backpropagating through a short lookahead window to keep multi-token edits fluent. We will trim the nuisance/causal discussion to 1–2 sentences and move details to the appendix.
>
> -----------
>
> >*While the paper presents fairly comprehensive results...*
>
> We appreciate the request to better interpret and evaluate the numerical result.
>     We will expand the text around Table 1 and add a short interpretation of the definitions from Section 6.1. We intended to measure Completeness - "did the targeted property change?", and Selectivity - "did unrelated properties remain stable?", and Reliability - the harmonic mean of the two. We will also add Pareto curve of completeness and selectivity to better interpret the results. As shown in Table 1, our method achieves consistently higher completeness and selectivity and, in most cases, higher reliability across datasets—all without any additional supervised‑probe training. Moreover, Appendix A.3 includes an ablation demonstrating that our margin‑loss objective is very effective for this results and outperforms the simpler alternatives.
>
> -------------
> > *Multi-token evaluation: While I find the multi-token lookahead method interesting...*
>
> LA‑HDMI performs inference‑time multi‑token lookahead edits within a user‑specified window. A single numeric ``quantitative'' evaluation is inherently dependent on each the input sequence because users deliberately choose how far the generated sequence can deviate from the proposed edits (to preserve coherency, being natural or fictional, etc) by adjusting the hyperparameters $\lambda_{\text{fact}}$ and $\alpha$.
>     Therefore there is a tradeoff between how coherent the text is and how successful the algorithm was to apply the edits which should be determined per instance by the users.
>     Moreover,  currently no standardized dataset for user‑specified multi‑token edits with ground‑truth labels exists; building such a benchmark is valuable but out of scope of this work.

---

> > ### Author Response · Authors · 2025-11-23
> >
> > >*Significance: My primary concern about the paper is that the authors do not...*
> >
> > Probing's goal is to predict which properties (such as sentiment or subject number) are represented by a model from hidden states. A major criticism
> > that applies to most existing work in probing is that being able to predict a property from hidden embeddings does not imply that the model indeed uses it for next token generation. Causal probing addresses this issue by performing interventions that remove or alter a property $Z$ in the representation and then measuring the downstream effects on the predictions [1][2][3].
> > HDMI provides a inference-time probe-free approach to test whether a targeted latent property $Z$ causally drives next-token predictions. By maximizing a logit-margin at a chosen step, we can flip specific continuations while quantifying completeness (did Z change?) and selectivity (were unrelated properties preserved?).
> >
> >
> >
> >
> > In practice, many causal probing methods rely on training a probe to intervene on the hidden state and find the direction along which they should steer the hidden state. For example, PGD [4] trains a probe to classify subject number from hidden states, and then uses the probe’s gradient to adjust the hidden state such that the perturbed state is classified by the probe with the opposite label (e.g., flipping the label from singular to plural).
> >
> > The problem is, this reliance on probes introduces extra property-specific supervision and training costs, since a separate probe must be trained for each property of interest. Moreover, there is a risk of misalignment in the sense that the probe imposes its own classification boundary on the hidden state which may not coincide with how the model internally encodes and uses a property for generating next tokens.
> > HDMI removes this dependency and uses model’s own output head as the readout for determining intervention direction.
> >
> > As an extension to HDMI, we introduced LA-HDMI, which adjusts token generation toward user-specified edits (e.g., agreement, gender, tense) while preserving fluency, without retraining. This is a lightweight alternative to PPLM/GeDi-style without the need for training an auxiliary discriminator, extra KL constraints, or probe training. LA‑HDMI does backpropagation through the softmax–expected-embedding–transition path to change the earlier hidden states for future edits.
> >
> > ### References
> >
> > >[1] Y. Elazar, S. Ravfogel, A. Jacovi, and Y. Goldberg, “Amnesic Probing: Behavioral Explanation with Amnesic Counterfactuals,” Transactions of the Association for Computational Linguistics, vol. 9, pp. 160–175, 2021. doi: 10.1162/tacl_a_00359
> >
> > >[2] M. Tucker, P. Qian, and R. Levy, “What If This Modified That? Syntactic Interventions with Counterfactual Embeddings,” in Findings of the Association for Computational Linguistics: ACL-IJCNLP 2021, pp. 862–875, 2021. doi: 10.18653/v1/2021.findings-acl.76
> >
> > >[3] A. Davies, J. Jiang, and C. Zhai, “Competence-Based Analysis of Language Models,” arXiv:2303.00333, 2023 (rev. 2024).
> >
> > >[4] A. Madry, A. Makelov, L. Schmidt, D. Tsipras, and A. Vladu, “Towards Deep Learning Models Resistant to Adversarial Attacks,” in International Conference on Learning Representations (ICLR), 2018. arXiv:1706.06083
> >
> >
> > -------------
> > Thank you once again for your feedback. We hope our responses have addressed your concerns and sincerely appreciate your consideration. If there are any additional questions or points that require clarification, please do not hesitate to let us know.

---

> ### Author Response · Authors · 2025-11-25
>
> Dear Reviewer,
>
> We would like to clarify that our core contribution—HDMI—already receives comprehensive quantitative evaluation on two benchmarks (16 datasets): the LGD agreement corpus and the CausalGym suites, each tested on two model families (Meta‑Llama‑3‑8B‑Instruct and Pythia‑70M). We report Completeness, Selectivity, and their harmonic mean (Reliability) following Canby et al. (2024), with results summarized in Table 1 and expanded in Appendix A.1. Across suites, HDMI delivers consistently strong Reliability versus other methods without training any probe to drive the intervention. We also include ablations and analyses: Appendix A.2 (Table~3) shows that replacing the margin objective with a target‑only objective substantially reduces Reliability on LGD.
>
> While the tables emphasize HDMI, the evaluation framework itself is not specific to HDMI; it applies to LA‑HDMI as well. At the token level, our two criteria—Completeness and Selectivity—align one‑to‑one with edit‑control metrics in other papers: Completeness corresponds to Edit Success Rate (ESR: fraction of specified edit positions realized exactly), and Selectivity corresponds to Non‑Target Stability (NTS: fraction of tokens outside the edit set unchanged relative to a matched base continuation). In our runs, we achieved near‑unit Completeness (ESR) at the edit position with near‑unit Selectivity (NTS) for non-edit positions.
>
> However, LLM‑judged metrics are not feasible here because there is no gold edited text (true label) for each input. Some generations may need to deviate from the original sentence to preserve fluency or factuality; conversely, forcing the local edits while keeping the rest of the sentence as close as possible to the original can degrade fluency or realism but may align with user intent. This should be specified by the user’s intent (fictional/factual/fluent). Given this inherent ambiguity, constructing a gold corpus or exhaustively obtaining human/LLM ratings for each prompt is not feasible and is beyond the scope of this work. We appreciate any suggestions the reviewer may have.

---

### Meta-Review · Area_Chair_kU9R · 2026-01-06

**Summary:**

1. The scope and significance of the proposed method is unclear.
2. Evaluation interpretation is weak, the metrics are unclear, and it requires better analysis or examples.
3. The evaluation for LA-HDMI (multi-token lookahead) is too qualitative and insufficient, and the reviewers ask for quantitative success and fluency metrics.
4. There is confusion between the proposed method and techniques in the existing literature. For example, the relationship between causal probing and causal mediation analysis is unclear.
5. There are concerns about the practical impact and general capability side effects (e.g., MMLU/GSM8K) and selectivity beyond probes.
6. The presentation is overly dense, and there is unnecessary formalism. Also, there lacks intuitive explanation.

**Reviewer Concerns:**

1. The concerns about the presentation and writing are addressed by the rebuttal.
2. The concerns about the evaluation of the proposed methods are partially addressed, and broad capability evaluation is not addressed.
3. The authors have clarified that HDMI is causal probing (intervene on hidden states to test causal influence on next-token behavior), distinct from correlational/linear probing. They position LA-HDMI as a use-case (fluency-preserving local edits) and promise to revise footnotes/text to avoid implying the paper is mainly “text editing.”
4. The debate about the relation between the proposed method and techniques in the existing literature remains.

**Reviewer Scores:**

1. Reviewer NBLB might increase their score to 5 or 6, because their concerns about presentation, motivation, and interpretation of metrics should have been addressed.
2. Reviewer G5nM's concerns are mainly about the comparison between the proposed method and the existing literature. It seems that extensive discussion would be required to address these concerns. Their score might be changed to 4 or 5.
3. Reviewer pf6F decreased their score during the discussion, and the concerns are largely unresolved even after many rounds of discussion. Thus their final score would be 3 or 4.
4. Reviewer uVoy would increase their score to 5.

---

### Decision · Program_Chairs · 2026-01-26

Reject